# Trends in the Consumption of Antidepressant Drugs before and during the COVID-19 Pandemic in the Canary Islands, Spain: The Case of the Province of Las Palmas

**DOI:** 10.3390/healthcare11101425

**Published:** 2023-05-15

**Authors:** Vanessa Moreno, Sandra Dévora, Susana Abdala-Kuri, Alexis Oliva

**Affiliations:** 1Department of Physical Medicine and Pharmacology, Faculty of Pharmacy, University of La Laguna, 38200 Tenerife, Spain; sdevora@ull.edu.es (S.D.); sabdala@ull.edu.es (S.A.-K.); 2Department of Chemical Engineering and Pharmaceutical Technolgy, Faculty of Pharmacy, University of La Laguna, 38200 Tenerife, Spain

**Keywords:** antidepressants, depression, COVID-19, consumption, defined daily dose, urban/rural, Spain, Canary Islands

## Abstract

The use of antidepressants (ADs) has increased significantly as a result of COVID-19 and its consequences. However, there are some notable differences in the relative levels of use between geographical areas and population groups. The aim of this work is to assess the impact of COVID-19 on the consumption of ADs in the Canary Islands, focusing on the islands of Gran Canaria, Fuerteventura and Lanzarote, by analyzing the trends in prescriptions of ADs during the pandemic period (2020) compared to the pre-pandemic period (2016–2020). Data were extracted from the community pharmacy wholesaler at a population level. Consumption patterns are expressed as the number of defined daily doses per 1000 inhabitant/day. The overall consumption of DIDs was higher in Gran Canaria, mainly in urban areas and the capital. It was similar in both Lanzarote and Fuerteventura, but particularly localized in the capital, which are considered semi-urban areas. Lanzarote and Fuerteventura present the same pattern of prescription ADs use, whereas Gran Canaria is notably different. This finding was also observed in the more consumed active pharmaceutical ingredients, although small inter-island variations in the ranking and percentages were observed. Sertraline and escitalopram are two of the most prescribed N06AB ADs, whereas the most recent N06AX ADs such as venlafaxine, mirtazapine and desvenlafaxine are more commonly prescribed. These differences in prescription ADs can be explained by demographical characteristics, population size, the fact of living in an urban area and general medical practice. In this context, the COVID-19 pandemic did not have an impact on the overall trend of the use of ADs between 2016 and 2020 in the islands under study.

## 1. Introduction

The new coronavirus disease (COVID-19) was first diagnosed in December 2019 in Wuhan City, China [1]. On 11 March, the emergence of the disease was classified as a pandemic by the World Health Organization (WHO), following a marked increase in cases around the world. The COVID-19 pandemic has profoundly affected the social, physical and mental well-being of individuals. Since the emergence of COVID-19, an increase in symptoms of anxiety, depression and stress has been observed in the general population and especially in healthcare workers [2,3,4]. In addition, social isolation, fear of infection and loss of loved ones, lack of income or work have all contributed to exacerbating mental health problems.

The Spanish Society of Psychiatry has said that depressive disorders could increase by up to 20% in the coming months and years because of COVID-19 and the deep social and economic crisis will affect the most vulnerable populations [5]. Recent studies conducted in many countries report a marked increase in the prevalence of depression symptoms in the population before and during the pandemic [6,7,8,9].

Given the significant rise in the prevalence of depression associated with COVID-19, the consumption of antidepressants (ADs) has also increased on a worldwide basis [6]. The sharp rise in the consumption of ADs is a major concern given the limited evidence on the long-term effectiveness and safety of ADs.

The newer antidepressants include selective serotonin reuptake inhibitors (SSRIs) and other antidepressants, with the following Anatomical Therapeutic Chemical (ATC) codes: N06AB and N06AX, respectively, whereas the older antidepressants include tricyclic antidepressants (TCAs) with the ATC code: N06AA. The recent increase in the use of SSRIs has been related to the increase in their long-term use as well as to the fact that they are increasingly being prescribed for conditions other than depression-related conditions [10]. At the same time, the incidence of SSRI use has decreased [11]. Different studies have reported that factors such as urbanicity level, the percentage of immigrants, gender, population over 65 years of age, education level, geographical area (capital and other areas outside the capital), prices, etc., may explain the possible differences in the use of newer and older ADs [10,12,13,14,15]. However, it is of interest to investigate the trends in AD use, especially, if the newer ADs are equally available and used at similar levels regardless of geographical area of residence or socioeconomic status [10].

The primary aim of the present study was to analyze prescription AD use in 2020 and compare this to a similar period in the four years preceding the pandemic (2016–2019) in the province of Las Palmas, Canary Islands, Spain. The second aim was to study the use of newer (N06AB and N06AX) and older ADs (N06AA) during the study period in the three islands that make up the province of Las Palmas. The third aim was to determine whether the possible differences in AD use may be related to the demographic and socioeconomic characteristics of each island as an example of local and isolated geographical areas. In order to do this, a descriptive and retrospective study of the use of ADs drugs was conducted, based on the methodology of the ATC classification and the defined daily dose (DDD) and the raw data obtained from the wholesalers who supply the community pharmacies at the population level were used as the database.

## 2. Materials and Methods

### 2.1. Data

A study was carried out on the use of ADs drugs included in the N06A therapeutic group according to the ATC classification in the out-of-hospital setting in the province of Las Palmas, Canary Islands, Spain (Figure 1).

The following four subgroups of ADs drugs were considered: (1) N06AA, non-selective monoamine reuptake inhibitors (or tricyclic antidepressants, TCAs); (2) N06AB, selective serotonin reuptake inhibitors (SSRIs); (3) N06AG, selective monoamine oxidase A inhibitor (MAOIs); (4) N06AX, other antidepressants. The dispensation of MAOIs was negligible (<0.01%) and only three subgroups were considered for this.

The present study used the raw data obtained from wholesalers who supply the community pharmacies at the population level in the province of Las Palmas during the period 2016–2020. The data were provided by the Pharmaceutical Cooperatives of the Canary Islands (COFARCA) and the Spanish Pharmaceutical Cooperative (COFARES). The data collected cover the entire distribution of the drug in the evaluated area, since there are no other supply channels that could interfere with the results obtained. Only data on ADs sold in the pharmacies under prescription were collected [16]. The data provided were the number of units sold of the different pharmaceutical preparations according to the ATC classification, date of sale, national code and pharmaceutical preparation name (active pharmaceutical ingredient (API), dose, strength and units). The postal code (ZIP) was also included, which provides information about the municipality where the community pharmacy is located, but not its identity, thus maintaining its anonymity in accordance with current data protection law in Spain [17]. In addition, two other databases were created: the first can be used to associate each ADs pharmaceutical preparation with the therapeutic subgroup, subgroup/API and DDD, and the second can be used to associate the postal code (ZIP) with a geographical area (province, island, municipality) and the population change during the analysis period. The combination of these three databases is the final aggregated database used in this study.

The prescription AD use for each API was expressed in population dose per day, which corresponds to the defined daily dose (DDD) per 1000 inhabitants/day (DID) [18].

Population data were obtained from a publicly accessible demographic database [19]. The database and dynamic tables of Microsoft Office Excel^®^ 2016 were used for the data processing. Data for Spain as a whole were provided by the Spanish Ministry of Health as dispensation data under prescription in community pharmacies [20].

### 2.2. Statistical Analysis

The aim of the study was to examine the variation of DID in function of the different geographical areas during the 2016–2020 period. The following linear model based on va-riance analysis (ANOVA) was used:(1)$$y_{ij}=\beta_{0}+\left(\beta_{1}+\beta_{3}\cdot X_{2}\right)\cdot X_{1}+\beta_{2}\cdot X_{2}+\epsilon_{ij}$$
where *y_ij_* is the dependent variable; the two independent variables used were: the time, *X*_1_, (in years) and the geographical area, *X*_2_, where *ϵ_ij_* is the residual random term assuming it is independent with normal distribution with zero mean and constant variance (N~(0, σ^2^)). The interaction “geographical area x time” was included to analyze the variation of the drug’s consumption in each geographical area with the time. In order to do this, the function lm () from R-program was used. Tukey test was used in the comparison between two levels using the function pairwise. *t*. test () from R-program [21]. This is the simpler model as it can be applied by using other variables. In addition, this model allows more independent variables and their corresponding interaction terms to be included.

### 2.3. Ethics Approval

The patient/subject’s data such as age, gender or pathology (i.e., primary data) were not provided and therefore, the ethics committee report of University of La Laguna or informed consent from patient/subject/were not required. The data analysis was conducted in accordance with relevant guidelines and regulations.

## 3. Results

### 3.1. Population and Demographic Data

Table 1 shows the population and demographic data at the provincial and island level. During the analyzed period, the population at the provincial level increased by 0.95%, with this increase being similar for Lanzarote and Fuerteventura (2.31% and 2.43%, respectively), whereas the increase in Gran Canaria was only 0.5%. Gran Canaria has a higher population density living in an urban area and 44.6% of the residents live in capital.

A similar situation was observed for the island of Lanzarote, 41.5% of the population live in the capital, Arrecife, with a population density of 2845 inhabitants per km^2^ (in 2020), although this data could be misleading since Arrecife is the smallest municipality in terms of surface area, whereas the mean average population density on the island is 184.2 inhabitants per km^2^. The island of Fuerteventura with double the surface area of Lanzarote, is the least densely populated, with 72.1 inhabitants per km^2^, and the smallest proportion of elderly people (11.11%). However, both of the latter two islands are considered as semi-urban areas [22] and their economy is based on tourism and related activities (≈80%) [23].

### 3.2. Analysis by Overall DID

Table 2 shows the percentages in the variation of the overall DID during the study period at the provincial level and for each island. The overall consumption of ADs was higher throughout 2020 in comparison to the estimated consumption in the period before the pandemic (approximately one point above the estimated value), except for Fuerteventura, where the consumption was practically the same, the difference was only 0.1%. Lanzarote experienced an increase of 3.45% in 2020 compared to 2019, although this data could be overestimated since the consumption in 2019 was lower than expected. At the provincial level, the increase was 2.34%, similar to that observed on Gran Canaria (2.45%).

However, the observed increases in DID for the year 2020 above the estimate are not significant. This is because the 95% prediction intervals for the estimate include the observed value. The results were 38.31 [35.92, 40.70] and 96.14 [92.46, 99.84] for Lanzarote and Gran Canaria, respectively.

The variability in overall DID was analyzed between the different geographical areas using a linear model to test for time trends. The null hypothesis for the geographical area variable was rejected (*p* < 0.05), and it was also rejected for the interaction term “geographical area x time”. This fact supposes that the overall DID varied between geographical areas during the period study as well as the overall DID variation rate (see Appendix A in Appendix A). The DID evolution for Lanzarote and Fuerteventura was similar during the period 2016–2020, with a consumption rate of 1.28 DID per year with 95% confidence intervals [0.756–1.80], whereas for Gran Canaria, the overall DID values were two and half times higher, with a consumption rate of 4.90 [3.24–6.15] DID per year. The level of dispensation (i.e., intercept) was higher in Gran Canaria (71.67 DID), while Lanzarote and Fuerteventura have similar values (30.92 vs. 28.27 DID). However, they are different because the 95% confidence limits for the means difference do not include the zero value (0.270, 5.097; *p* = 0.0365). This differences in consumption could be related with the demographic and socioeconomic characteristics of each island. At the same time, the overall DID values at a nationwide level (i.e., Spain as a whole) are higher than the overall DID in Gran Canaria, by approximately 10 points and twice the value of those on Lanzarote and Fuerteventura. However, the national consumption rate was 2.75 [1.57–3.98] DID per year, approximately half that of Gran Canaria, but twice that of the other two islands.

The second step was to analyze the variation of the overall DID per quarter and year in all the islands to determine whether there are differences in DID variation, especially in 2020 due to COVID-19. This allows us to see possible changes in the trend over a short period of time. At first, the data analysis confirms the results obtained using the annual consumption data, with the consumption rates being similar (see Appendix A in Appendix A). Figure 2 shows the prescription trends of ADs per quarter and year during the period 2016–2020 for each island together with the 95% prediction bands according to the proposed model. A similar trend was observed in all islands during the 2020 year. A peak dispensing of ADs was observed in the first quarter, followed by a sharp decrease in the second quarter, which corresponds to the lockdown declared by the Spanish government (15 March to 21 June 2020). When the lockdown ended, the DID values then remained more stable until the end of year. A detailed analysis shows the possible presence of an outlier in Lanzarote and Gran Canaria, localized at the first quarter of the 2020 year. To detect this, we used the Grubbs test from the “outliers” package in R for outlier detection. The results show that the *p*-value was higher than 0.05 in both cases, which means that we do not have enough evidence to reject the null hypothesis (i.e., Ho: there is no outlier in the data) and therefore, the maximum value is not an outlier.

The prescription rate would be expected to increase with the rise in the number of aged people (expressed as a percentage of over 65-year olds in the population). The results show that there is a significant correlation between the variation of overall DID and the percentage of aged people in all the islands (r^2^ > 0.99). However, the inclusion of this variable in the proposed model did not result in any statistical significance (*p* = 0.898). In addition, the comparison of both models through an F-test indicate that the simpler model provides a better fit for the data.

### 3.3. Analysis by Antidepressants Drug and Classes

To the best of the authors’ knowledge, the geographical differences in the use of different AD classes have not been examined previously in such detail and over such a long study period. Table 3 shows the mean consumption (expressed as percentages) of different AD subgroups for each island and in Spain as a whole during the study period.

The N06AB subgroup (i.e., SSRI) was the most consumed, regardless of the island, with a mean contribution of 60.92%, followed by the N06AX subgroup (i.e., other ADs), with 34.76%, and finally the N06AA subgroup (i.e., TCAs) with 4.1% in the study period. These values are similar to those observed at the provincial level.

An ANOVA was used to analyze the variability associated with the dispensation of different ADs subgroups and the possible differences between islands during the study period, considering the island and the time as independent variables according to the proposed model in Section 2.

Firstly, the coefficient of adjusted determination (R^2^) was 0.989 and thus, the proposed model is suitable for the data interpretation. The null hypothesis for the island variable was accepted (*p* > 0.05), whereas it was rejected for the interaction “island x time”. This fact supposes that the trends of different ADs subgroups do not change during the study period within each island, but there are differences between islands. Tukey’s test shows that there are no differences between Lanzarote and Fuerteventura (*p* > 0.05), whereas Gran Canaria is different (*p* < 0.01). In order to confirm this result, a second ana-lysis was performed to determine the possible differences between islands for each therapeutic subgroup and to determine their evolution during this period (Table 4).

The results in Table 4 show that the consumption of TCAs have remained fairly stable in all the islands over the last five years, although the dispensation level was different, 5.65% on average in Fuerteventura versus 3.15% for Gran Canaria. The SSRIs were the most consumed medication, although with a negative tendency, whereas the “other antidepressants” subgroup showed a positive tendency. However, the level of dispensation (i.e., intercept), and consumption rate of both subgroups depends on each island. In this regard, there are no differences between Fuerteventura and Lanzarote with respect to the consumption rate of SSRIs, with a decrease of 2.56% per year being observed, although the rate was lower in Gran Canaria (−1.63% per year), whereas in the “other antidepressants” group (N06AX) consumption increased by the same magnitude (+1.62% per year). In graphical terms, the consumption rate of the latter subgroup is different for Lanzarote and Fuerteventura (see Figure 3). However, both are statistically equal since the 95% confidence interval for the mean difference between both islands included the zero value (Table 4), although this data should be confirmed in the future since the probability level obtained was slightly higher than 5% (*p* = 0.0656). The data in Spain as a whole shows the same change in tendency but at a lower rate, −0.86 [−1.03, −0.695] DID per year, three times lower than those observed in Lanzarote and Fuerteventura [20].

The next step was to examine the trends in the prescription of AD drugs at a provincial level to determine whether there are differences in 2020 compared to 2019. Table 5 shows the variation in DID for each active pharmaceutical ingredient (API) and therapeutic subgroup with respect to the annual DID. Sertraline ranks first, followed by escitalopram and desvenlafaxine, although fluoxetine is close behind. It should be noted that, of the first four APIs, three belong to the SSRI while the fourth corresponds to the N06AX subgroup. All of these account for 47% of the total DID, although a decrease of 1.8% was observed in 2020.

The analysis by API and subgroup identified amitriptyline as the most consumed medication within the TCAs at 93.8% in 2019, decreasing slightly in 2020 to 93.4%. Clomipramine is a long way behind in second place at 3.85% in 2019, but decreasing to 3.66% in 2020. There are six APIs in subgroup N06AB, with fluvoxamine being the least demanded, accounting for only 0.5% of the subgroup’s total DID. Sertraline is the most consumed API, accounting for 30.36% of the total DID, reaching 31% in 2020, followed by escitalopram at 25.9%, decreasing to 25.15% in 2020. Finally, fluoxetine and citalopram account for 13.26% and 11.11%, respectively, and their consumption has remained stable over the analyzed period.

The N06AX subgroup is the largest, with a total of eleven APIs. Desvenlafaxine ranks first at 19.4%, followed by mirtazapine, venlafaxine, duloxetine and trazodone with percentages ranging from 17.20% to 15.43%. The remaining APIs have values below 3%, with the exception of vortioxetine which has a contribution of 9.0% in 2019, rising to 10.92% in 2020. All APIs present slight variations (±0.5 DID on average) in consumption compared to 2020, except for vortioxetine which shows an increase close to 1 DID.

## 4. Discussion

Gran Canaria and Lanzarote present overall annual DID values above the expected one in 2020, according to the expected evolution of DID in the period before the pandemic, whereas Fuerteventura data are within the expected value. However, the observed values in DID for all islands in 2020 are within the 95% prediction intervals for the estimated value, which contains the observed value. At first, the variation of population during the 2020 year and socioeconomic characteristics could explained this result. In addition, the data analysis by quarter and year show sharp changes in trends during the first two quarters of the 2020 year (see Figure 2), more pronounced being in Lanzarote and Gran Canaria, where the lockdown and associated measures taken during the pandemic could explain the changes in trends in the consumption of ADs. Regarding this point, the populations of both Lanzarote and Fuerteventura depend mainly on tourism, which was one of the most heavily affected sectors by the COVID-19. Despite financial aid, the sector suffered important consequences, especially the workers; as such, the expected prescription of ADs should be higher. In addition, the number of tourists who visited both islands was close to 5 million in 2019 compared to 1.3 million in 2020, a decrease of 72% [23]. In order to evaluate the possible impact of the COVID 19 and its consequences on the consumption of ADs, especially, in the first quarter of 2020 in both Gran Canaria and Lanzarote, where it DID values are outside the 95% prediction intervals, we analyzed the presence of outliers using the Grubb’s test. The results confirmed that there were no outliers in either case. The observed differences fall within the range of variability derived from the proposed model, with no unusual causes to explain the findings. This result implies that the COVID-19 pandemic did not change the overall trend per quarter and year in the period 2016–2020 in the analyzed islands.

The geographical and demographic differences may reflect discrepancies in the use of different AD classes due to prescription practice, access to mental health care services, quality of care, level of urbanicity, percentage of over 65-year olds in the population, education and socioeconomic level, etc. [10,24,25,26,27,28]. Bauer et al. [29] analyzed the prescription patterns of antidepressants and the factors influencing depression in twelve European countries. In sum, the prescribing of antidepressants varies by country, and the type of antidepressant chosen is influenced by the physician—as well as patient-related factors. Morrison et al. [15] analyzed factors influencing variation in the prescription of ADs in Scotland. These authors report that almost half of the variation can be explained by population and general medical practice.

In this context, the results obtained seem to indicate that the increase was higher in Gran Canaria, mainly in urban areas, and in Lanzarote, but localized in the capital, whereas it was not significant on Fuerteventura, considered as semi-urban area. Morrison et al. [15] reported that the dispensing of ADs in Scotland was higher in urban areas than in rural areas, differing to that described by other authors [12]. These results show differences between islands with different levels of urbanity, but it is necessary to investigate whether consumption is higher in capital areas than in noncapital areas. In such a situation, an analysis by municipalities may be helpful in the confirmation of the authors’ previous results.

With respect to the type of antidepressant, SSRIs were the most commonly used, followed by the “other antidepressant”, whereas the consumption of TCAs remained stable during the study period. This pattern was observed on all the islands. In addition, this result is consistent with other national data and comparable with those reported in other European countries [12,24,25,26,27,28].

The data expressed in DID provide accurate information on the use of prescription ADs, but the problem could be analyzed from another point of view, for example, the distribution of each subgroup, expressed as a percentage of overall DID, which allows its evolution in a geographical area to be known over time. This procedure allows us to normalize the data and comparisons between the islands.

A decrease in SSRI consumption was observed from the year 2016, but at the same time an increase of the same magnitude was found in the N06AX subgroup, which was higher in Lanzarote and Fuerteventura (2.56% DID per year) compared to Gran Canaria (1.63% DID per year), whereas the consumption of TCAs was low and stable (see Table 4), with values of 3.22% in Gran Canaria and slightly higher ones in Lanzarote (4.24%) and Fuerteventura (5.48%). The data obtained suggest that Lanzarote and Fuerteventura present the same pattern of AD prescription, similar to those observed in Gran Canaria, although with a different level of prescription and consumption rate. At the nationwide level, the data show that, on average, almost 62% of prescriptions were for SSRIs compared to 58.3% at the provincial level, whereas 34.5% of prescriptions were for the “other antidepressant” subgroup, four points below the value at the provincial level. At the same time, the consumption rate of SSRIs and other antidepressants have the same change in tendency as those observed on the islands, but at a lower rate (0.86% DID per year). The authors believe that the geographical area of residence does not play a significant role in AD prescription practices, but the population characteristics and size could explain this difference.

Among the different classes of ADs, SSRI and TCA were, in absolute terms, the two most prescribed classes in 2020 compared to 2019, whereas the consumption of N06AX subgroup was 10.12% versus 8.23% in 2019 at the provincial level (Table 5). Different studies have investigated the dispensation of AD drugs in regions of Spain and European countries. For example, González-López et al. [12] estimated the consumption rate for SSRIs and the other ADs group in seventeen basic health areas in the province of Almeria, Andalusia, Spain, during the period 2004–2010. However, the results showed a high variability in the dispensation trends for both AD subgroups among health areas. Nevertheless, these data cannot be used in the comparative analysis in the present study since only two subgroups were included and the data analysis was not performed at a provincial level. In all the studies, the main conclusion was a rise in the dispensation of the SSRIs and other AD groups, while there was a reduction in the dispensation of TCAs [12,24,27,30,31]. In this respect, SSRIs have practically replaced TCAs because of their greater tolerability and safety.

However, the analysis of SSRIs shows a rise in the consumption of sertraline, escitalopram, paroxetine, fluoxetine, and citalopram, this last one without any change in dispensation and finally, fluvoxamine the least used (Table 5). The low consumption of fluvoxamine may be related to the higher incidence of adverse gastrointestinal effects than that observed with other antidepressants and its higher cost, especially when compared to fluoxetine [32]. All these APIs are indicated to treat major depressive disorder, anxiety and social anxiety, whose symptoms were exacerbated during the pandemic. This result is in line with those obtained in other regions of Spain and European countries, although the ranking changes slightly. Studies conducted in the United Kingdom and the Netherlands show that escitalopram is the most consumed, while paroxetine consumption has stabilized or declined [24,30]. Sertraline consumption doubled during the pandemic in the United Kingdom [30,31].

In the “other antidepressant” group, the most recent medications such as desvenlafaxine, mirtazapine, venlafaxine and duloxetine were the most consumed, and trazadone was found among the classic N06AX ADs. However, the greatest increase was found in the case of venlafaxine and mirtazapine, the most recent ones, followed by trazadone, among the classic N06AX ADs, whereas the consumption of duloxetine and desvenlafaxine decreased slightly (Table 5). A reduction in the dispensation of mianserine and reboxetine was also observed, although these drugs are practically not dispensed for different reasons [33,34]. Vortioxetine should also be highlighted because its consumption increased by 29.6% in 2020 compared to 2019, which was a smaller increase than the increase of 38% observed between 2018 and 2019. This marked increase is due to its recent introduction in the market, in 2016, and to its high tolerance, safety and action on cognitive deficits present in major depressive disorders compared to other antidepressants [35].

Amitriptyline, whose consumption accounts for more than 90% of total TCA consumption, had an increase of 4.22% in its use in 2020, similar to the pre-pandemic period. The data obtained seem to indicate a stable consumption on all the islands since 2016, the year when the present study period started. This trend has also been observed in other European countries and Spanish regions [24,25,26]. However, nationwide data show a rise in consumption, but a very small one, with an increase of barely 0.073 [0.0669–0.0791] DID per year [20].

Furthermore, the variations observed for the different APIs did not differ significantly with respect to previous years, with the exception of vortioxetine, on all the islands. The ranking of different AD categories was fairly stable on each island and year, although the percentages and ranking varied slightly. For example, on the island of Lanzarote, escitalopram was the most consumed AD followed by sertraline, whereas mirtazapine and duloxetine were the most consumed on Fuerteventura in the N06AX subgroup. In this context, both islands are characterized by specialist doctors located in the capital who initiate the prescription according to the pathology. The authors believe that this factor plays an important role in determining ADs prescription rates. However, Gran Canaria has two reference hospital located in the capital where the patients from all over the island are referred to depending on their place of residence and not their illness.

The dispensation of ADs for conditions other than depression, such as anxiety disorders, neuropathic pain (TCs) or for smoking cessation (bupropion), as well as the improved detection of depression by family physicians, the availability of safer and more effective novel drugs, are some of the factors involved in the increase in AD consumption. In addition, new clinical indications have appeared because different effects on cognitive function require ADs at doses substantially lower than those for depression and these new indications have also contributed to this increase [36]. It is unlikely that the COVID-19 epidemic will have contributed to increased prescription AD use on the islands studied by 2020, although the potential contribution is difficult to quantify. One of the limitations of the present study is the lack of information in the database on the indications which the ADs were prescribed for as well as the fact that the patients’ demographic data (age and gender) were not provided. Thus, the results are based on dispensing and not prescription data, so the reason for the diagnosis is unknown. In addition, the potential bias is difficult to quantify since the prescription is associated to postcode (ZIP) and not the township where the patients live [18]. In this regard, the DID data at the island and provincial level are more accurate since they included all the population who live in that zone. However, several strengths include the analysis of consumption based on data on ADs sold in the pharmacies under prescription, and the length of the time period of the study. Although prescription ADs use has been studied in various countries in the last twenty years, few studies in Europe have focused on local and isolated zones such as islands to establish a pattern of use of prescription ADs, which is a novel approach.

## 5. Conclusions

The overall annual consumption of ADs was higher throughout 2020 when compared to the estimated value in pre-pandemic period, although this was noteworthy in Gran Canaria, mainly in urban areas and in Lanzarote, but localized in the capital (≈7.87%). However, these differences are within the 95% prediction intervals according to the model proposed. The variation in population size and socioeconomic characteristic could justify this observation. In addition, the COVID-19 pandemic did not have an impact on the overall trend of the use of ADs between 2016 and 2020 in the islands under study. Lanzarote and Fuerteventura show the same pattern of prescription ADs use, whereas Gran Canaria is significantly different. However, this pattern has not changed during the period 2016–2020. This finding was also observed between the most consumed APIs, although small variations in the ranking and percentages were observed between the islands. Sertraline and escitalopram are two of the most prescribed SSRIs, whereas the most recent N06AX antidepressants such as venlafaxine, mirtazapine and desvenlafaxine are currently more commonly prescribed. The data analysis in Spain as a whole shows the same trends in prescriptions for SSRIs and other antidepressants, whereas TCAs have seen a slight upturn in their consumption in the last five years.

These differences in the use of prescription ADs can be explained by demographic characteristics, population size, the fact of living in an urban area and general medical practice. It is unlikely that the COVID-19 epidemic will have contributed to an increase in the use of prescription ADs on the islands by 2020, although its potential contribution will be the subject of future research over a longer time period.

## Figures and Tables

**Figure 1 healthcare-11-01425-f001:**
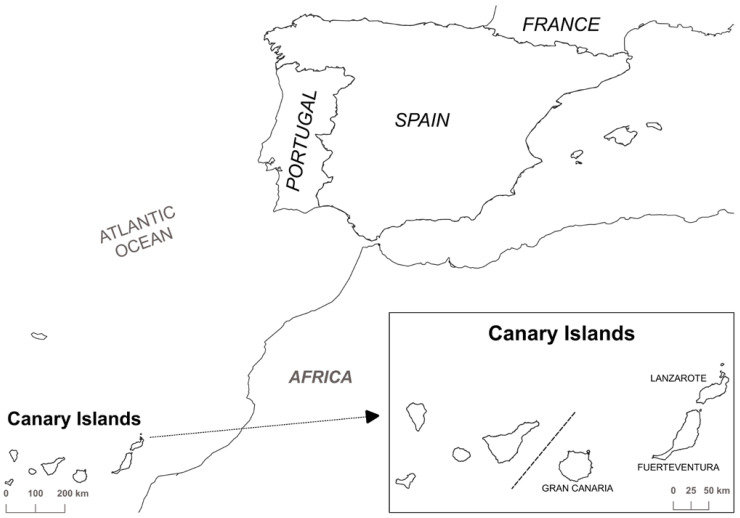
Maps of the Canary Islands, Spain, with the three islands that make up the province of Las Palmas.

**Figure 2 healthcare-11-01425-f002:**
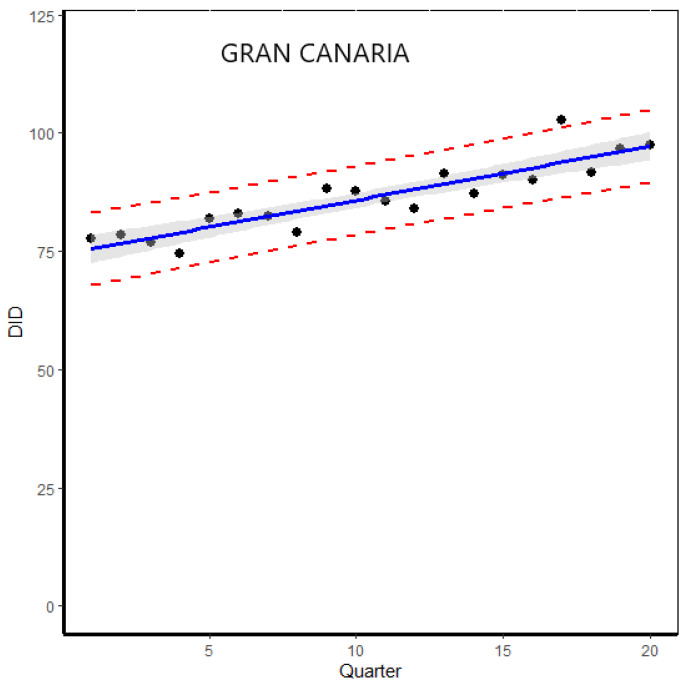
Variation of the overall DID per quarter and year for each island during the period 2016–2020 (in a total 20 quarter) together with the 95% prediction bands (dashed red line) and the estimated regression line (in blue) with 95% confidence limits (grey shadow area).

**Figure 3 healthcare-11-01425-f003:**
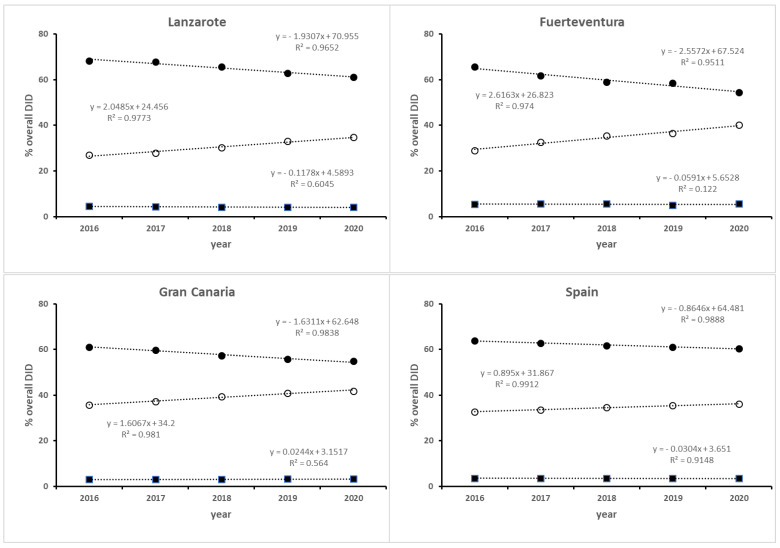
The consumption rate of different therapeutic subgroups (expressed as a percentage of the overall DID) at island level and Spain as a whole. SSRI (●); TCAs (■); Other ADs (○).

**Table 1 healthcare-11-01425-t001:** Demographic data and administrative distribution of the province of Las Palmas. The population density (inhabitants per km^2^) and over-65-year-olds (%) corresponds to the year of 2020. %∆ = Percentage change in population over the analyzed period.

			Population				Population Density	Over- 65 Years (%)
Island	2016	2017	2018	2019	2020	%∆	2020	2020
Gran Canaria	845,195	843,158	846,717	851,231	855,521	0.5	548.4	16.97
Lanzarote	145,084	147,023	149,183	152,289	155,812	2.31	184.2	12.57
Fuerteventura	107,521	110,299	113,275	116,886	119,732	2.43	72.1	11.11
Province	1,097,800	1,100,480	1,109,175	1,120,406	1,131,065	0.95	278.2	13.07

**Table 2 healthcare-11-01425-t002:** Percentages in the variation of the overall DID at different levels (provincial, island and Spain as a whole). The value with an asterisk (*) is the DID estimated for 2020 based on the pre-pandemic period (2016–2019) using the proposed model.

Year	Province	Gran Canaria	Lanzarote	Fuerteventura	Spain
2016	66.39	76.84	32.36	30.15	75.47
2017	70.06	81.54	34.17	30.11	77.21
2018	73.92	86.34	35.43	31.76	79.91
2019	76.80	90.00	36.00	33.84	83.07
2020	82.45	97.08	38.83	34.67	86.28
2020 *	81.70	96.14	38.31	34.66	85.88

**Table 3 healthcare-11-01425-t003:** Variation of DID (expressed as a percentage of overall DID) for each therapeutic subgroup of ADs during the period 2016–2020 at different levels (provincial, island and Spain as a whole). The data corresponds to the mean and standard deviation.

Subgroup	Lanzarote	Fuerteventura	Gran Canaria	Province	Spain
N06AA	4.24 ± 0.24	5.48 ± 0.27	3.22 ± 0.05	3.39 ± 0.04	3.56 ± 0.05
N06AB	65.16 ± 3.11	59.85 ± 4.14	57.76 ± 2.60	58.32 ± 2.69	61.89 ± 1.37
N06AX	30.60 ± 3.28	34.67 ± 4.19	39.02 ± 2.56	38.28 ± 2.67	34.55 ± 1.42

**Table 4 healthcare-11-01425-t004:** Results of ANOVA for the factors of therapeutic subgroup and island according to the proposed model. (^1^) Data refer to Fuerteventura (reference level). (*) Significative for *p* < 0.05.

Subgroup	Coefficients	Estimate	2.5%	97.5%	Pr(>|t|)
N06AA (TCAs)	Intercept ^1^	5.65	5.19	6.12	<0.01 *
	Year ^1^	−0.059	−0.200	0.081	**0.366**
	Gran Canaria	−2.50	−3.16	−1.84	<0.01 *
	Lanzarote	−1.06	−1.72	−0.40	<0.01 *
	year × Gran Canaria	0.083	−0.11	0.28	**0.366**
	year × Lanzarote	−0.059	−0.26	0.14	**0.520**
N06AB (SSRIs)	Intercept ^1^	67.52	65.73	69.32	<0.01 *
	Year ^1^	−2.56	−3.10	−2.02	<0.01 *
	Gran Canaria	−4.88	−7.41	−2.34	<0.01 *
	Lanzarote	3.43	0.89	5.97	0.014 *
	year × Gran Canaria	0.926	0.16	1.69	0.0229
	year × Lanzarote	0.626	−0.14	1.39	**0.0969**
N06AX (other ADs)	Intercept ^1^	26.82	25.38	28.26	<0.01 *
	Year ^1^	2.62	2.18	3.05	<0.01 *
	Gran Canaria	7.38	5.34	9.41	<0.01 *
	Lanzarote	−2.37	−4.40	−0.33	0.0271 *
	year × Gran Canaria	−1.01	−1.62	−0.40	<0.01 *
	year × Lanzarote	−0.568	−1.18	0.045	**0.0656**

**Table 5 healthcare-11-01425-t005:** Variation of overall DID by subgroup/API during the period 2018–2020, before and during the pandemic.

Subgroup/API	2018	2019	2020	∆ (%) 2018–2019	∆ (%) 2019–2020
N06AA	2.46	2.60	2.85	5.69	9.61
N06AA02-Imipramine	0.01	0.01	0.01	0	0.00
N06AA04-Clomipramine	0.11	0.10	0.11	9.1	10.00
N06AA09-Amitriptiline	2.28	2.44	2.68	7.20	9.84
N06AA10-Nortriptiline	0.01	0.01	0.01	0.00	0.00
N06AA12-Doxepin	0.04	0.04	0.04	0.00	0.00
N06AA21-Maprotiline	0.02	0.01	0.00	−50	−100
N06AB	42.89	43.28	45.56	0.91	5.27
N06AB03-Fluoxetine	5.73	5.74	5.88	0.17	2.44
N06AB04-Citalopram	4.81	4.81	4.98	0.00	3.53
N06AB05-Paroxetine	8.20	8.14	8.89	0.73	9.21
N06AB06-Sertraline	12.69	13.14	14.11	3.55	7.38
N06AB08-Fluvoxamine	0.24	0.23	0.21	−4.17	0.00
N06AB10-Escitalopram	11.22	11.21	11.48	0.01	2.41
N06AX	28.57	30.92	34.05	8.23	10.12
N06AX03-Mianserin	0.22	0.02	0.02	−90.9	0.00
N06AX05-Trazodona	4.52	4.77	5.26	5.53	10.27
N06AX11-Mirtazapini	5.13	5.32	5.75	3.70	8.08
N06AX12-Bupropion	0.61	0.60	0.63	1.64	5.00
N06AX14-Tianeptine	0.19	0.34	0.37	78.95	8.82
N06AX16-Venlafaxine	5.15	5.21	5.65	1.17	8.44
N06AX18-Reboxetine	0.08	0.07	0.06	−12.50	−14.28
N06AX21-Duloxetine	4.47	4.81	5.08	7.61	5.61
N06AX22-Agomelatine	0.95	0.99	0.84	4.21	−15.15
N06AX23-Desvenlafaxine	5.24	6.00	6.76	14.50	12.67
N06AX26-Vortioxetine	2.02	2.80	3.63	38.61	29.64
Overall DID	73.92	76.80	82.45	3.90%	7.35%

## Data Availability

The data that support the findings of this study are available from COFARES and COFARCA, but restrictions apply to the availability of these data, which were used under license for the current study, and so are not publicly available. Data are however available from the corresponding author upon reasonable request and with permission of COFARES and COFARCA.

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
