# Peer review of "Trends in the Consumption of Antidepressant Drugs before and during the COVID-19 Pandemic in the Canary Islands, Spain: The Case of the Province of Las Palmas"

_healthcare, 2023, doi:10.3390/healthcare11101425_

Round 1
Reviewer 1 Report (Previous Reviewer 1)
The authors have addressed all my comments.
I only suggest editing or removing Supplementary Figure 1.
Author Response
Reviewer #1
The authors have addressed all my comments.
Thank you for your words.
I only suggest editing or removing Supplementary Figure 1.
In this context, we have removed Supplementary Figure 1 from the revised version.
Reviewer 2 Report (Previous Reviewer 2)
I have already made comments, imo the article is good to be published as it is now and the Editor should not have rejected the paper in the first place.
It's good.
Author Response
Reviewer #2
I have already made comments, imo the article is good to be published as it is now and the Editor should not have rejected the paper in the first place.
Thank you for your words.
This manuscript is a resubmission of an earlier submission. The following is a list of the peer review reports and author responses from that submission.
Round 1
Reviewer 1 Report
In this article, the authors investigated antidepressant prescription (ADs) trends during the COVID-19 pandemic in 2020 in comparison to the pre-pandemic period (2016-2019) to assess the impact of COVID-19 on ADs consumption in the Canary Islands. They found an increase in the overall consumption of ADs in 2020 in comparison to the pre-pandemic period, with a higher increase in urban areas. Sertraline and escitalopram were among the most prescribed N06AB ADs, whereas venlafaxine, mirtazapine and desvenlafaxine were among the the most recent commonly prescribed N06AX ADs.
In my opinion, this is a well-written article. I only have minor comments that can be useful to improve the manuscript before publication in Healthcare.
- - When describing the aims of the study at the end of the Introduction, a better rationale for the selection of these geographic areas, in terms of local and isolated zones such as islands, should be added.
- In paragraph “2. Materials and Methods”, the authors reported that they used “raw data” obtained from wholesalers. Which kind of raw data did they refer to?
- - In paragraph “2.2 Statistical analyses” the authors reported “yij is the dependent variable (i.e., DID response)”. What does DID response mean? The authors did not investigate the response rate of ADs and, in my opinion, talking about DID response should be misunderstanding.
- - Does Table 2 refer to percentage in the variation of the overall DID? If so, this should be clearly indicated in the legend of the table.
Author Response
REPLIES TO REVIEWERS´ COMMENTS
Reviewer#1
In this article, the authors investigated antidepressant prescription (ADs) trends during the COVID-19 pandemic in 2020 in comparison to the pre-pandemic period (2016-2019) to assess the impact of COVID-19 on ADs consumption in the Canary Islands. They found an increase in the overall consumption of ADs in 2020 in comparison to the pre-pandemic period, with a higher increase in urban areas. Sertraline and escitalopram were among the most prescribed N06AB ADs, whereas venlafaxine, mirtazapine and desvenlafaxine were among the most recent commonly prescribed N06AX ADs.
In my opinion, this is a well-written article. I only have minor comments that can be useful to improve the manuscript before publication in Healthcare.
- When describing the aims of the study at the end of the Introduction, a better rationale for the selection of these geographic areas, in terms of local and isolated zones such as islands, should be added.
Thank you for your suggestion. The phrase was added.
- In paragraph “2. Materials and Methods”, the authors reported that they used “raw data” obtained from wholesalers. Which kind of raw data did they refer to?
It is true; we have used raw data from the wholesalers (the list of data is shown in the text, lines: 117-120), which have to be modified in order to create the initial database. For example, the format of the 'date of sale' is different for each supplier (i.e. cooperative) and we have to make the appropriate change in order to get the same format. In addition, two other databases were created: the first allows us to associate each pharmaceutical preparation with the therapeutic subgroup, subgroup/API and DDD, and the second allows us to associate the postal code (ZIP) with a geographical area (province, island, municipality) and the population change during the analysis period. Combining these three databases, we obtain the final aggregated database used in this study. An example is provided in the supplementary material. (See Supplementary Figure S1). This last issue has been included in the revised version.
- In paragraph “2.2 Statistical analyses” the authors reported “yij is the dependent variable (i.e., DID response)”. What does DID response mean? The authors did not investigate the response rate of ADs and, in my opinion, talking about DID response should be misunderstanding.
It is true, the expression (i.e., DID response) was deleted.
- Does Table 2 refer to percentage in the variation of the overall DID? If so, this should be clearly indicated in the legend of the table.
Thank you for your suggestion. This change has been included.
Reviewer 2 Report
This is an interesting paper and appraoch. Congratulations. Few short comments below.
1. Abstract is not sufficient. Background focuses on explaining the aim of the study but doesn't give any real background.
2. The whole structure of the introduction seems a little messy, you have some kind of introduction and then you jump to conclusions and parts that should be in the discussion (third paragraph). Fourth paragraph should be finished with justification of your study more clearly.
3. Methodolgy seems sufficient but lines 116-117 - where else can Spanish people get their medication than from a pharmacy store? Do you mean illegal sources or...? If only from a pharmacy store, then this line doesn't seem neccesary and just cause, to some extent of course, misunderstanding.
3. Some parts of results should be moved to discussion as well, like agonizing areal data (157-165). Please work on creating results section that presents raw data you extracted. Rest goes into discussion.
4. Very interesting discussion, with upgrades from above points it will be great.
5. Conclusions are supported by presented data however, they should present the summary of what can be concluded, not what your data does not support (468).
Last but not least - the manuscript should undergo English editing. Please ask your native collegue to see it, there's not much to change but some parts could use editing to look more professional.
Author Response
REPLIES TO REVIEWERS´ COMMENTS
Reviewer #2
This is an interesting paper and approach. Congratulations. Few short comments below.
- Abstract is not sufficient. Background focuses on explaining the aim of the study but doesn't give any real background.
Thank you for your suggestion. A brief background has been included.
- The whole structure of the introduction seems a little messy, you have some kind of introduction and then you jump to conclusions and parts that should be in the discussion (third paragraph). Fourth paragraph should be finished with justification of your study more clearly.
Thank you for your suggestions. Both paragraphs were moved to the discussion. A summary has been added.
- Methodology seems sufficient but lines 116-117 - where else can Spanish people get their medication than from a pharmacy store? Do you mean illegal sources or...? If only from a pharmacy store, then this line doesn't seem necessary and just cause, to some extent of course, misunderstanding.
The Spanish people only can buy ADs in the community pharmacies under prescription (i.e., extra-hospital data). The lines 116-117 have been changed.
- Some parts of results should be moved to discussion as well, like agonizing areal data (157-165). Please work on creating results section that presents raw data you extracted. Rest goes into discussion.
The Results section was revised following your recommendation and some parts of results were moved to discussion.
- Very interesting discussion, with upgrades from above points it will be great.
Thank you for your words.
- Conclusions are supported by presented data however, they should present the summary of what can be concluded, not what your data does not support (468).
The conclusion section was revised based on your suggestions.
Last but not least - the manuscript should undergo English editing. Please ask your native collegue to see it, there's not much to change but some parts could use editing to look more professional.
Thank you for your suggestion.